

# Anxiolytic and anti-stress effects of acute administration of acetyl-L-carnitine in zebrafish

Lais Pancotto[1,*], Ricieri Mocelin[2,*], Matheus Marcon[2], Ana P. Herrmann[1] and Angelo Piato[1,2,3]

[1] Programa de Pós-Graduação em Farmacologia e Terapêutica, Universidade Federal do Rio Grande do Sul, Porto Alegre, RS, Brazil
[2] Programa de Pós-Graduação em Neurociências, Universidade Federal do Rio Grande do Sul, Porto Alegre, RS, Brazil
[3] Zebrafish Neuroscience Research Consortium (ZNRC), Los Angeles, United States of America
[*] These authors contributed equally to this work.

Corresponding author
Angelo Piato, angelopiato@ufrgs.br

## ABSTRACT

Studies have suggested that oxidative stress may contribute to the pathogenesis of mental disorders. In this context, molecules with antioxidant activity may be promising agents in the treatment of these deleterious conditions. Acetyl-L-carnitine (ALC) is a multi-target molecule that modulates the uptake of acetyl-CoA into the mitochondria during fatty acid oxidation, acetylcholine production, protein, and membrane phospholipid synthesis, capable of promoting neurogenesis in case of neuronal death. Moreover, neurochemical effects of ALC include modulation of brain energy and synaptic transmission of multiple neurotransmitters, including expression of type 2 metabotropic glutamate (mGlu2) receptors. The aim of this study was to investigate the effects of ALC in zebrafish by examining behavioral and biochemical parameters relevant to anxiety and mood disorders in zebrafish. ALC presented anxiolytic effects in both novel tank and light/dark tests and prevented the anxiety-like behavior induced by an acute stressor (net chasing). Furthermore, ALC was able to prevent the lipid peroxidation induced by acute stress in the zebrafish brain. The data presented here warrant further investigation of ALC as a potential agent in the treatment of neuropsychiatric disorders. Its good tolerability also subsidizes the additional studies necessary to assess its therapeutic potential in clinical settings.

## INTRODUCTION

Acetyl-L-carnitine (ALC) facilitates the movement of acetyl-CoA into the mitochondria during the oxidation of fatty acids in mammals (*Chapela et al., 2009*). Moreover, this molecule is widely consumed as a dietary supplement for physical exercise (*Ribas, Vargas & Wajner, 2014*; *Nicassio et al., 2017*). Recently, preclinical and clinical studies have demonstrated the effects of ALC on parameters relevant to anxiety, schizophrenia, and mood disorders; with onset of action faster than antidepressant drug and exert

neuroprotective, neurotrophic, and analgesic effects (*Levine et al., 2005*; *Wang et al., 2015*; *Traina, 2016*; *Singh et al., 2017*; *Nasca et al., 2017*; *Chiechio, Canonico & Grilli, 2017*).

A growing body of evidence suggests that psychiatric disorders such as anxiety and depression are associated with oxidative damage (*Ortiz et al., 2017*; *Niedzielska et al., 2016*; *Schiavone, Colaianna & Curtis, 2015*; *Cobb & Cole, 2015*; *Ng et al., 2008*), since a decrease in antioxidant capacity can impair the organism's protection against reactive oxygen species and cause damage to fatty acids, proteins, and DNA (*Maes et al., 2011*). Superoxide and hydroxyl radical (free radicals) or hydrogen peroxide and their derivatives (non-radical molecules) called reactive oxygen species (ROS) are responsible for causing oxidative damage (*Smaga et al., 2015*). The antioxidant defense mechanism they are the non-enzymatic (i.g. glutathione) and enzymatic antioxidants (i.g. superoxide dismutase and catalase) which show a trend to decrease in neuropsychiatric diseases (*Ozcan et al., 2004*; *Hassan et al., 2016*). Preclinical and clinical research has evaluated antioxidant compounds (i.g. N-acetylcysteine, resveratrol and curcumin) in the treatment of psychiatric disorders, and it has been reported that these compounds are able to protect against oxidative stress-induced neuronal damage, preventing lipid peroxidation and behavioral changes (*Mecocci & Polidori, 2012*; *Berk et al., 2014*; *Wang et al., 2014*; *Mocelin et al., 2015*; *Patel, 2016*; *Santos et al., 2017*).

With simple, rapid and cheaper tests when compared with rodents, zebrafish have been used as a powerful complementary model for the study of a variety of neuropsychiatric diseases through behavioral and biochemical parameters (*Stewart et al., 2015*; *Mocelin et al., 2015*; *Marcon et al., 2016*; *Marcon et al., 2018*; *Khan et al., 2017*). There are several behavioral protocols extensively used and described for this species, such as the novel tank and light/dark tests. The novel tank diving test is based on an anti-predatory defense mechanism that induces fish to swim at the bottom of the tank, whereas the light/dark test evaluates anxiety based on the innate preference of adult zebrafish to dark over light areas (*Levin, Bencan & Cerutti, 2007*; *Gebauer et al., 2011*; *Khan et al., 2017*; *Pittman & Piato, 2017*).

In addition to its role in lipid metabolism, ALC also possesses free radical scavenging properties, and may thus protect the cells from oxidative damage by acting as an antioxidant (*Gülçin, 2006*; *Sepand et al., 2016*). Therefore, the aim of this study was to investigate the effects of ALC in zebrafish by examining behavioral and biochemical parameters relevant to anxiety and mood disorders in zebrafish.

## MATERIALS AND METHODS
### Animals
A total of 240 adult zebrafish (*Danio rerio,* F. Hamilton 1822) wild-type short fin strain (6-month-old, 3–4 cm long) 50:50 male/female ratio were purchased from Delphis aquariums (Porto Alegre, Brazil). The fish were kept for 15 days in a closed acclimation tank system of 16 L (40 × 20 × 24 cm) identical to the experimental tanks. Housing conditions consisted only of a tank with water, heater, filter and aeration system, and were maintained as previously described in *Marcon et al. (2016)*. The tanks contained non-chlorinated, aerated

tap water (pH 7.0 $\pm$ 0.3; temperature 26 $\pm$ 1 °C; total ammonia at <0.01 mg/L; nitrite <0.01 mg/L; dissolved oxygen at 7.0 $\pm$ 0.4 mg/L; alkalinity at 22 mg/L CaCO$_3$ and total hardness at 5.8 mg/L), with a light/dark cycle of 14/10 h (lights on at 06:00 am). The fish were fed twice a day with a commercial flake fish food (Alcon BASIC®; Alcon, São Paulo, Brazil). On the experimental days, all the fish were only fed early in the morning before behavioral testing began. The order of testing was counterbalanced so that fasting time was randomized across experimental groups. All experiments were approved by the Ethics Committee of Universidade Federal do Rio Grande do Sul (#30992/2015).

## Drug and experimental design

O-Acetyl-L-carnitine hydrochloride (ALC, CAS number 5080-50-2) was acquired from Sigma-Aldrich (St Louis, Missouri, USA). In all experimental protocols (novel tank, light/dark, and acute chasing stress tests), the animals were treated or not with ALC (0.1, 1.0 and 10.0 mg/L) in a beaker for 10 min. In the first protocol, immediately after the treatment, the animals were placed in the novel tank test (NTT) for 6 min. In the second protocol, after the treatments, the animals were placed in the light/dark test (LDT) for 5 min. Finally, in the third protocol, the animals were treated as previously and then chased with a net for 2 min. Then, the animals were placed in the NTT. The biochemical analyses were performed in animals submitted to this last protocol. A control group was submitted to the same experimental conditions (stressed or not) but without treatment. Different sets of animals were used in each experimental protocol. The experimental design is shown in Fig. 1 and was based on the previously published study by Mocelin et al. (2015). All behavioral tests were performed between 09:00 am and 16:00 pm. The researchers who performed the behavioral tests and analyzed the data were unaware of the allocation of animals to the experimental groups. The concentrations were based on previous studies with another antioxidant compound (N-acetylcysteine) and pilot studies with a wider concentration range. The same concentrations were used in a chronic study with ALC (M Marcon, R Mocelin, A Araujo, A Herrmann & A Piato, 2018, unpublished data). We do not attempt to extrapolate the drug concentrations we used in a fish study to human dosage since there is not a straightforward calculation to be done. Since the half-life and other pharmacokinetic parameters of ALC in zebrafish are not known, it is difficult to precisely compare the concentration range that we observed here with the dose range for humans.

## Novel tank test (NTT)

The novel tank test followed the protocol already described in Mocelin et al. (2015). Briefly, the animals were separately moved to the apparatus (2.7-L tank, 24 × 8 × 20 cm, virtually divided into three equal horizontal zones and filled with standard tank water up to 15 cm) and video recorded for 6 min to be later analyzed by the ANY-maze™ software (Stoelting Co., Wood Dale, IL, USA). To evaluate exploratory behavior and locomotion we measured the parameters: total distance moved (m), number of transitions between zones, time spent in the upper and bottom zones of the tank, and number of transitions to the upper zone. Total distance and crossings were used as an indicator of overall locomotor activity. The

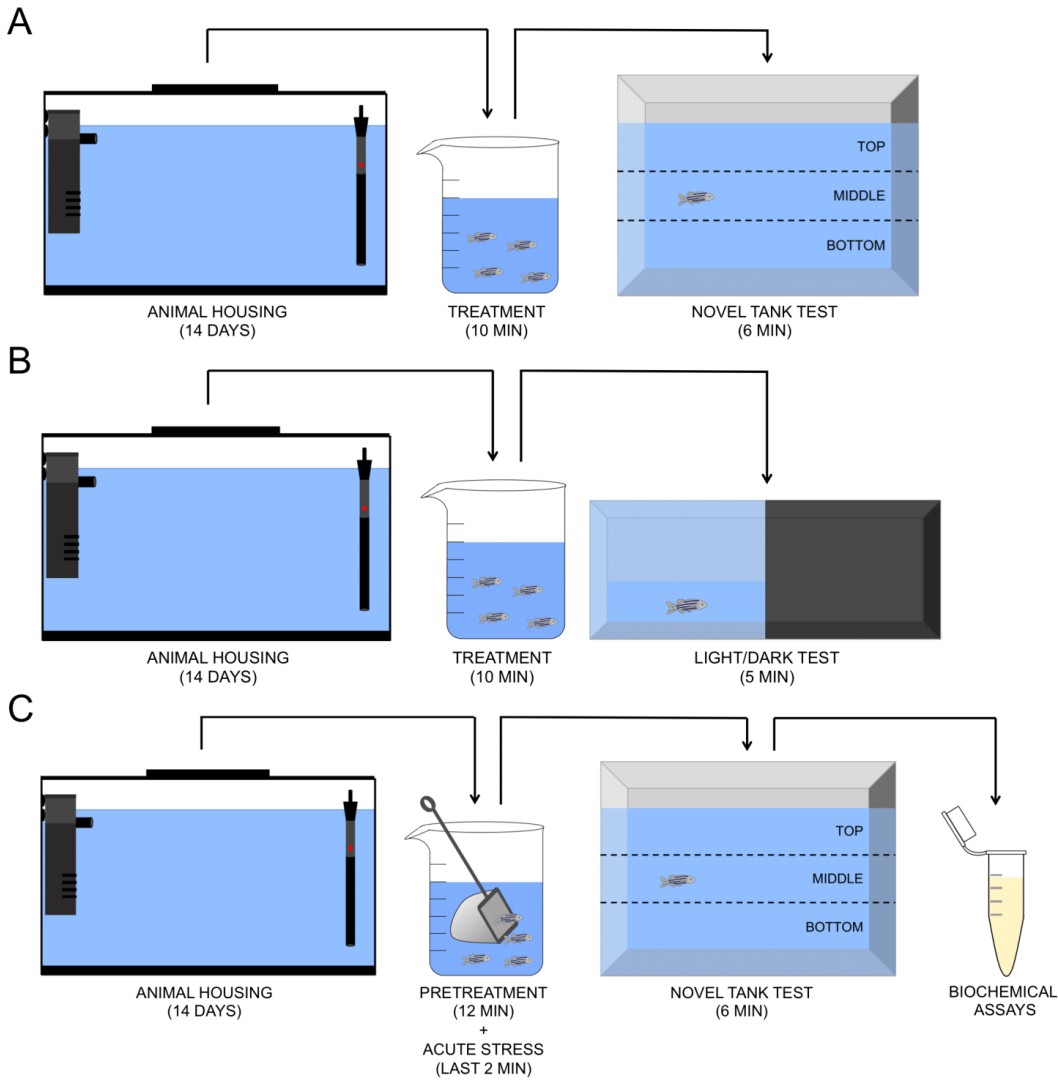

**Figure 1** Schematic representation of the experimental protocol. Novel tank test (A), light/dark test (B), and acute chasing stress and biochemical assays (C).

upper zone of the tank corresponds in rats and mice protocols to the periphery region of the open-field test. Alterations in time spent and number of crossings to this zone are frequently used as a parameter of anxiety in zebrafish (*Mocelin et al., 2015*; *Giacomini et al., 2016*; *Marcon et al., 2016*; *Mocelin et al., 2017*; *Marcon et al., 2018*).

## Light/dark test (LDT)

The light/dark test followed the protocol already reported by *Gebauer et al. (2011)*. Specifically, the apparatus consisted of a glass tank (18 × 9 × 7 cm) divided by a raised glass into a dark and a white compartment of equal sizes, with the water level set at 3 cm and the partition raised 1 cm above the tank floor. One fish at a time was positioned in the white zone of the apparatus immediately after treatment. We recorded the number of

crossings and the time spent in the white compartment for 5 min. Zebrafish have a natural preference for dark environments and the white compartment is very anxiogenic for this species; anxiolytics increase the time spent in the white compartment (*Maximino et al., 2010*; *Mocelin et al., 2015*).

### Acute chasing stress test (ACS)

The acute stress protocol was performed according to the previous study published by *Mocelin et al. (2015)*. Briefly, the animals were treated for 12 min and then chased for the last 2 min with a net before being moved to the novel tank, where they were recorded for 6 min. The behavioral parameters were quantified as described above for the NTT.

### Tissue preparation

Samples were collected and prepared as previously reported by *Mocelin et al. (2018)*. Specifically, after the ACS fish were anesthetized by immersion in cold water and euthanized by decapitation. Each independent sample was then obtained by pooling four brains, which were homogenized on ice in 600 μL phosphate buffered saline (PBS, pH 7.4; Sigma-Aldrich, St. Louis, MO, USA). The homogenate was centrifuged at 10,000 g for 10 min at 4 °C in a cooling centrifuge, and the supernatant was packed in microtubes for further assays.

### Protein determination

Protein was determined by the Coomassie blue method described in detail by Bradford (1976). Specifically, we used bovine serum albumin (Sigma-Aldrich, St. Louis, MO, USA) as standard and the absorbance of samples was measured at 595 nm.

### Lipid peroxidation (TBARS)

Lipid peroxidation was measured by the quantification of thiobarbituric acid reactive species (TBARS) production according to the method reported by (*Draper & Hadley, 1990*). More specifically, we followed the protocol described by *Mocelin et al. (2018)*, in which 50 μL of the sample (80–100 μg protein) was mixed with 75 μL of trichloroacetic acid (TCA 10%; Sigma-Aldrich, St. Louis, MO, USA) and centrifuged at 6,000 rpm for 5 min at 4 °C in a cooling centrifuge. In the supernatants were added to 75 μL thiobarbituric acid (TBA 0.67%; Sigma-Aldrich, St. Louis, MO, USA), then homogenized in a vortex for 5s and heated at 100 °C for 30 min. TBARS levels were measured by absorbance (532 nm) in a microplate reader, using malondialdehyde (MDA; Sigma-Aldrich, St. Louis, MO, USA) as a standard, and results were expressed as nmol MDA/mg protein.

### Reduced thiol (SH) and Non-protein thiols levels (NPSH)

SH and NPSH levels were determined and measured at 412 nm in a microplate reader according to the method described by *Ellman (1959)*. More specifically, we followed the steps described by *Mocelin et al. (2018)*. Briefly, for SH the samples (60–80 μg protein) were added to 10 mM 5,5-dithio-bis-2-nitrobenzoic acid (DTNB) dissolved in ethanol, developing yellow color after 1 h. The NPSH were similarly assessed, except that the sample was mixed with equal volumes of the 10% trichloroacetic acid (TCA) and centrifuged (6,000 rpm, 5 min). The supernatant was used for the biochemical assay. Results were expressed as μmol SH/mg protein.

### Superoxide dismutase (SOD) and catalase (CAT) activities

SOD and CAT activities were determined according to the method reported by *Misra & Fridovich (1972)* and *Aebi (1984)*, respectively. The protocol followed the more specific details described by *Dal Santo et al. (2014)*. Specifically, SOD activity was quantified in a microplate reader (480 nm) by testing the inhibition of radical superoxide reaction of the sample (20–30 $\mu$g protein) in the presence of adrenalin, monitoring adrenochrome formation in a medium containing a glycine-NaOH buffer (pH 10) and adrenaline (1 mM). CAT activity was assessed by measuring the decrease in $H_2O_2$ absorbance in a microplate reader (240 nm). The assay mixture consisted of sample (20–30 $\mu$g protein), phosphate buffered saline (pH 7.4), and 5 $\mu$L $H_2O_2$(0.3 M). Results were expressed as units/mg protein.

### Statistics

Normality and homogeneity of variance of the data were checked by D'Agostino-Pearson and Levene tests, respectively. Results were analyzed by one- or two-way ANOVA followed by Tukey's post hoc test. Two-way ANOVA was used to identify the main effects of stress and treatment, as well as their interactions. Data are expressed as a mean + standard error of the mean (S.E.M.). The level of significance was set at $p < 0.05$.

## RESULTS

### Behavioral parameters

Figure 2 shows the effects of ALC (0.1, 1.0 and 10.0 mg/L) on the novel tank test in zebrafish. ALC significantly increased the time spent in the top (0.1 and 1.0 mg/L, Fig. 2D) and decreased the time spent the bottom (0.1 mg/L, Fig. 2E) zone of the tank ($F_{3,77} = 8.0$, $p = 0.0001$ and $F_{3,77} = 5.6$, $p = 0.0016$, respectively). Locomotor parameters of groups treated with ALC (0.1, 1.0, and 10.0 mg/L) did not differ from control (Figs. 2A and 2B).

In the light/dark test, ALC (0.1 and 10.0 mg/L) significantly increased the time spent in the lit side of the tank when compared to control ($F_{3,92} = 3.6$, $p = 0.0161$, Fig. 3B). The number of crossings between the light and dark compartments was not altered by any of the concentrations ($F_{3,92} = 0.9$, $p = 0.4284$, Fig. 3A).

Figure 4 shows the effects of ALC in the acute chasing stress (ACS) in zebrafish and Table 1 summarizes the two-way ANOVA analysis. As expected, ACS decreased the distance total traveled, crossings, entries and time in the top area (Figs. 4A–4D, respectively) and increased the time in the bottom area (Fig. 4E). ALC (0.1, 1.0 and 10.0 mg/L) prevented the effects of ACS on the time in the top and bottom areas in the novel tank test (Figs. 4D and 4E ). Also, ALC (0.1 mg/L) prevented the effects of ACS on the total distance traveled.

### Biochemical parameters

Figure 5 shows the effects of ALC (0.1, 1.0 and 10.0 mg/L) on oxidative status. ACS significantly increased lipid peroxidation (TBARS), non-protein sulfhydryl (NPSH) and superoxide dismutase (SOD) activity (Figs. 5A, 5C and 5D, respectively), but did not alter sulfhydryl (SH) content and catalase (CAT) activity (Figs. 5B and 5E, respectively). Treatment with ALC (0.1, 1.0 and 10.0 mg/L) prevented oxidative damage as measured by

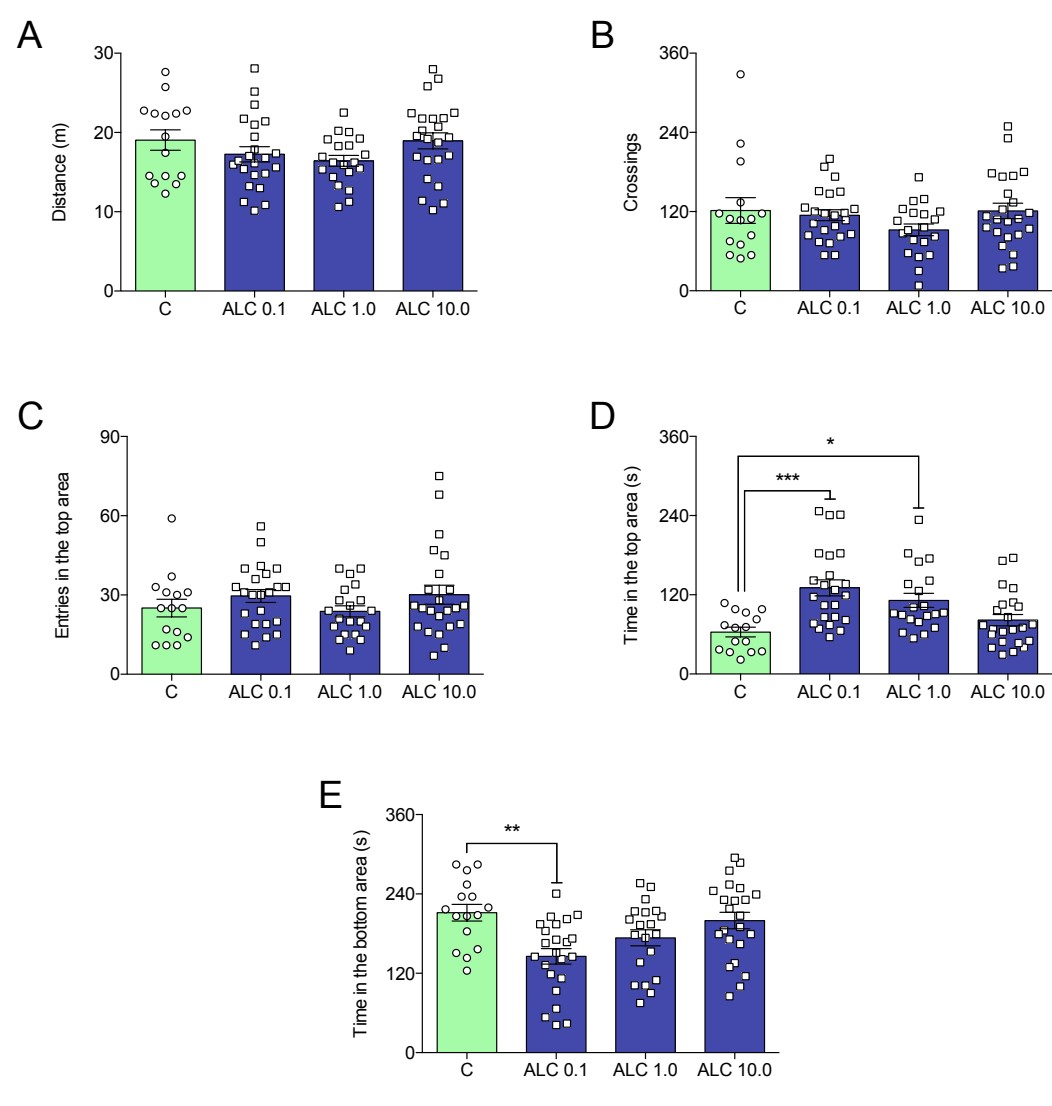

**Figure 2** **Effects of ALC (0.1, 1.0 and 10.0 mg/L) behavioral parameters in zebrafish submitted to the novel tank test.** (A) distance traveled, (B) number of crossings, (C) entries and (D) time in the upper zone, and (E) time in the bottom zone. The data are presented as the mean + S.E.M. One-way ANOVA followed by Tukey post hoc test. $n = 15$–$23$. * $p < 0.05$, ** $p < 0.01$, *** $p < 0.001$ vs. control group.

TBARS. ALC also prevented the increase of antioxidant defenses as measured by NPSH (0.1 mg/L) and SOD (0.1, 1.0 and 10.0 mg/L). Two-way ANOVA analyses were summarized in Table 2.

## DISCUSSION

Here, we showed for the first time that ALC presents anxiolytic effects in both novel tank and light/dark tests in zebrafish. Moreover, ALC was able to prevent the anxiogenic effects and lipid peroxidation induced by an acute stress protocol. These results indicate a potential use of ALC in mental disorders related to stress.

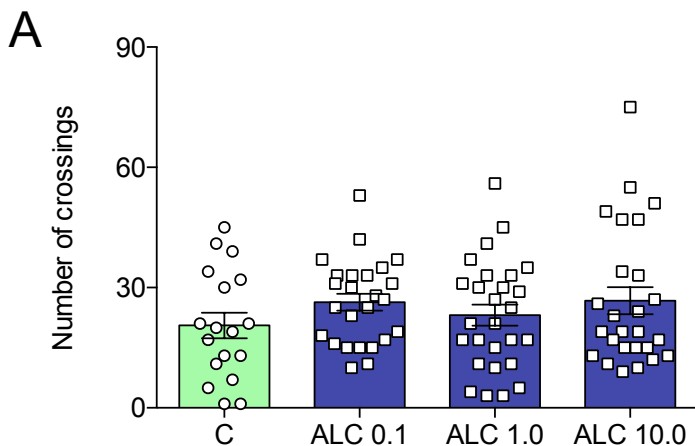

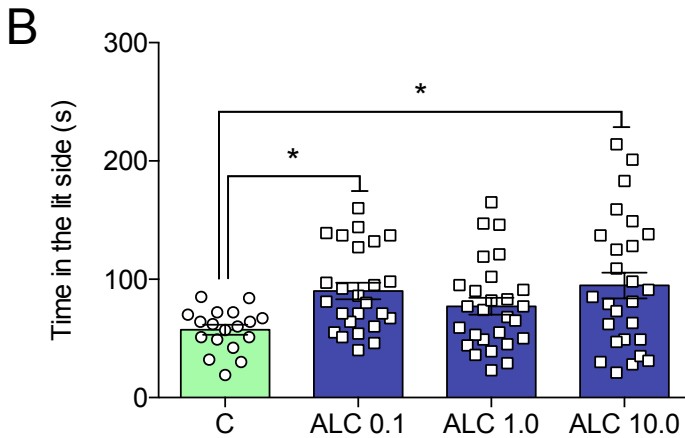

**Figure 3** **Effects of ALC (0.1, 1.0 and 10.0 mg/L) in the light/dark test in zebrafish.** (A) number of crossings and (B) time in the lit side. The data are presented as the mean + S.E.M. One-way ANOVA followed by Tukey post hoc test. $n = 18$–$27$. *$p < 0.05$ vs. control group.

ALC increased the time spent in the upper as well as decreased the time spent in the bottom zones of the tank. Previous studies have shown that anxiolytic drugs such as buspirone, fluoxetine, diazepam, and ethanol increase the time spent in this zone (*Bencan, Sledge & Levin, 2009*; *Egan et al., 2009*; *Gebauer et al., 2011*). In the light/dark test, ALC increased the time spent in the lit side of the tank. This effect was observed with other drugs as clonazepam, bromazepam, diazepam, buspirone, and ethanol (*Gebauer et al., 2011*). Additionally, multi-target drugs other than ALC, for instance, N-acetylcysteine (NAC) and taurine, also increase the time in the lit side in the LTD in zebrafish (*Mocelin et al., 2015*; *Mezzomo et al., 2015*). In both NTT and LDT, ALC presented biphasic response. We can only speculate that different mechanisms of action may be involved in the effects of low versus high dose, but lower and higher concentrations would have to be tested for us to have a bigger picture of the dose–response relationship. ALC also prevented the locomotor impairment and anxiogenic behavior induced by the chasing stress protocol. Recently, our

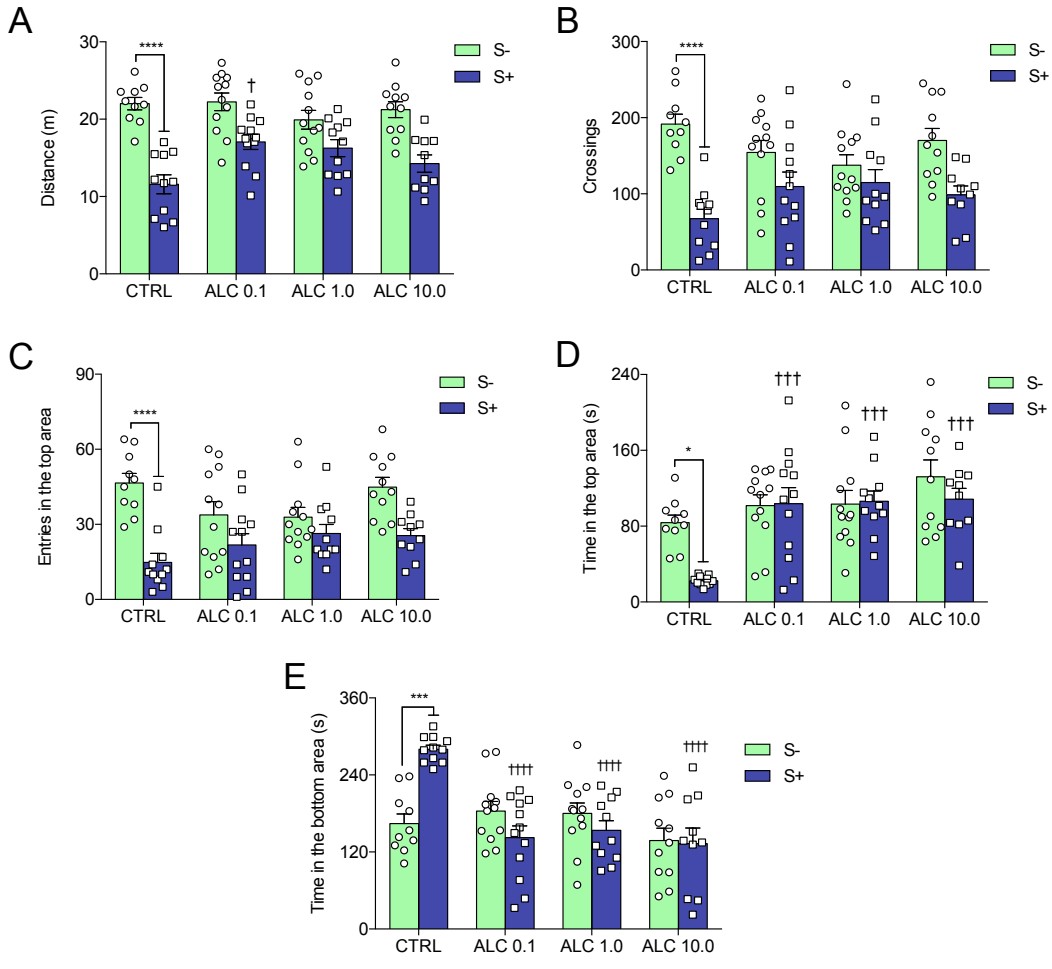

**Figure 4** **Effects of ALC pretreatment against stress-induced changes in behavioral parameters in zebrafish.** (A) distance traveled, (B) number of crossings, (C) entries and (D) time in the upper zone, and (E) time in the bottom zone. The data are presented as the mean + S.E.M. Two-way ANOVA followed by Bonferroni's test. $n = 10$–$12$. $^{*}p < 0.05$, $^{***}p < 0.001$, $^{****}p < 0.0001$ vs. control group (S−); $^{\dagger}p < 0.05$, $^{\dagger\dagger\dagger}p < 0.001$, $^{\dagger\dagger\dagger\dagger}p < 0.0001$ vs. stressed control group (S+).

group has shown that fluoxetine, diazepam, and NAC prevented the effects of a similar stress protocol in zebrafish (*Mocelin et al., 2015*; *Giacomini et al., 2016*).

The anxiolytic and antidepressant effects of ALC have been already reported in rodents (*Levine et al., 2005*; *Wang et al., 2015*; *Lau et al., 2017*). ALC modulates the cholinergic system by increasing acetyl-CoA content and choline acetyltransferase activity. Moreover, it modulates GABAergic, dopaminergic and glutamatergic neurotransmitter systems (*Chapela et al., 2009*; *Nasca et al., 2013*; *Wang et al., 2014*; *Singh et al., 2016*; *Chiechio, Canonico & Grilli, 2017*). In rats, ALC decreased the immobility time in the forced swim test and increased sucrose preference in 3 days of treatment, whereas 14 days were necessary to obtain the same effects with clomipramine (*Nasca et al., 2013*).

**Table 1  Results of two-way analysis of variance (ANOVA) of behavioral analysis and the interaction between treatment with ALC and acute chasing stress.**

| Dependent variable | Effects | *F*-value | DF | *P*-value |
|---|---|---|---|---|
| Total distance | Interaction | 3.46 | 3,81 | **0.0201** |
| | ALC | 2.39 | 3,81 | 0.0745 |
| | Stress | 71.34 | 3,81 | **0.0001** |
| Crossings | Interaction | 4.04 | 3,81 | **0.0099** |
| | ALC | 0.10 | 3,81 | 0.9583 |
| | Stress | 37.11 | 3,81 | **0.0001** |
| Entries in the top | Interaction | 3.47 | 3,81 | **0.0198** |
| | ALC | 1.18 | 3,81 | 0.3215 |
| | Stress | 36.10 | 3,81 | **0.0001** |
| Time in the top | Interaction | 2.72 | 3,81 | **0.0499** |
| | ALC | 9.81 | 3,81 | **0.0001** |
| | Stress | 4.86 | 3,81 | **0.0303** |
| Time in the bottom | Interaction | 9.02 | 3,81 | **0.0001** |
| | ALC | 9.03 | 3,81 | **0.0001** |
| | Stress | 0.84 | 3,81 | 0.3613 |

**Notes.**
DF, degrees of freedom.
Significant effects ($p < 0.05$) are given in bold.

Under normal conditions, damage by reactive oxygen species (ROS) is kept in control by efficient antioxidant systems, such as SOD and CAT enzymes, as well as non-enzymatic scavengers (*Schiavone et al., 2013*; *Schiavone, Colaianna & Curtis, 2015*; *Sandi & Haller, 2015*). Studies have demonstrated that ALC protects cells against lipid peroxidation and membrane breakdown through hydrogen peroxide scavenging (*Kumaran et al., 2003*; *Gülçin, 2006*), and can promote the expression of antioxidant enzymes such as SOD and CAT (*Augustyniak & Skrzydlewska, 2010*; *Li et al., 2012*).

Even though the ACS protocol increased antioxidant defenses (NPSH and SOD), it also caused lipid peroxidation (TBARS), which may indicate a possible adaptive response to ROS production during stressful conditions. Similar results were observed in zebrafish and reported in a previous study from our group using acute restraint stress (*Dal Santo et al., 2014*). Even though detection of MDA levels by HPLC would be a more specific indicator of lipid peroxidation, the TBARS assay we used in this study has been reported by many previous articles using samples from zebrafish and other animals (*Mihara & Uchiyama, 1978*; *Sunderman et al., 1985*; *Armstrong & Browne, 1994*; *Yagi, 1998*; *Kim et al., 2011*; *Basu et al., 2014*; *Yavuzer et al., 2016*). The association of these factors could be related to the prevented effects of ALC, and that our results indicate a deficit in antioxidant defenses against lipid peroxidation in zebrafish submitted to the ACS protocol, providing

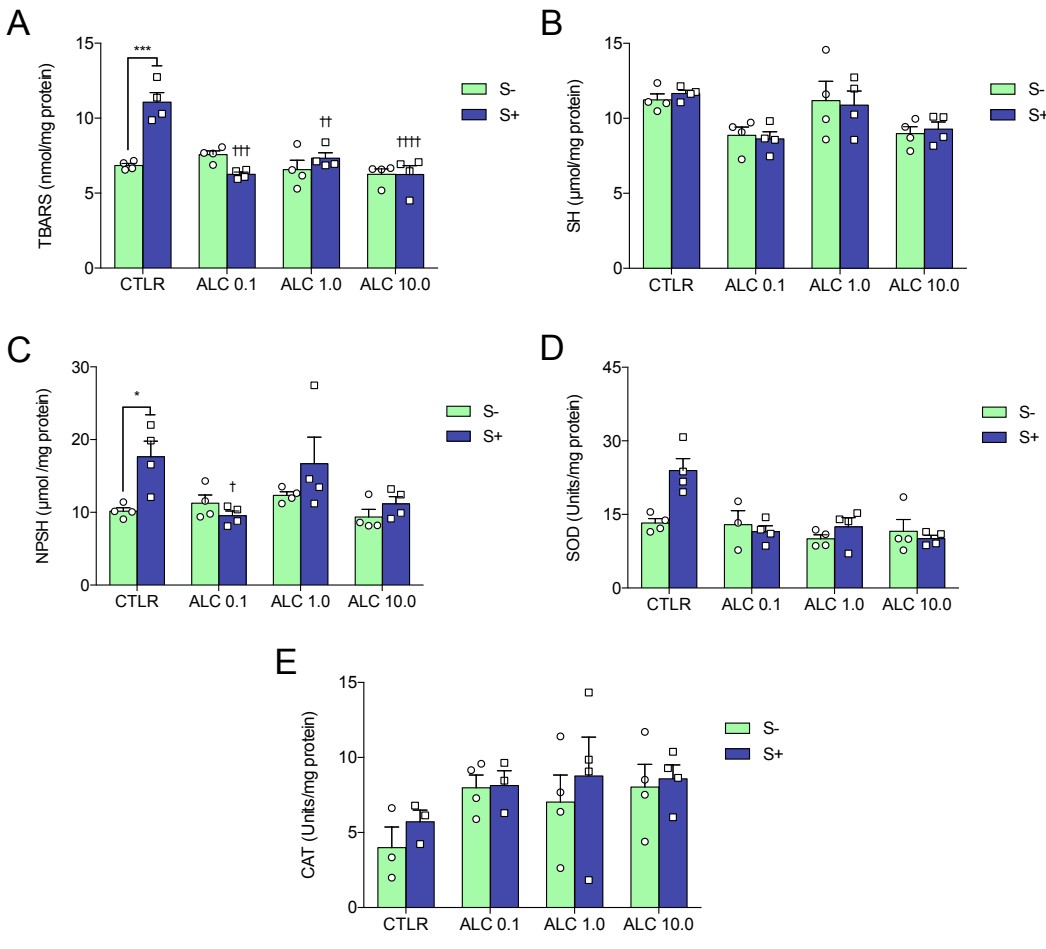

**Figure 5** **Effects of ALC pretreatment against stress-induced changes in biochemical parameters in zebrafish.** (A) thiobarbituric acid reactive substances, (B) sulfhydryl, (C) non-protein sulphydryl, (D) superoxide dismutase, and (E) catalase. The data are presented as the mean + S.E.M. Two-way ANOVA followed by Bonferroni's test. $n = 3$–$4$. $^*p < 0.05$, $^{***}p < 0.001$ vs. control group (S−); $^{\dagger}p < 0.05$, $^{\dagger\dagger\dagger}p < 0.001$, $^{\dagger\dagger\dagger\dagger}p < 0.0001$ vs. stressed control group (S+).

further evidence for the hypothesis of an association between behavior and ROS with the pathophysiology of mental disorders stress-related and their prevention by ALC.

## CONCLUSION

ALC is already widely used as supplementation for people who want to lose weight/fat burner, but only a few studies assessed its effects on stress-related outcomes. In addition to its antioxidant actions, ALC is also able to restore mitochondrial function, which is relevant to combat the dysregulation of fatty acid metabolism in the mitochondria-associated with psychiatric disorders. Furthermore, there is evidence that ALC increases expression of metabotropic glutamate receptors via epigenetic mechanisms (*Nasca et al., 2013*), which is also relevant for the pathophysiology of depression and other stress-related disorders.

**Table 2 Results of two-way analysis of variance (ANOVA) of biochemical analysis and the interaction between treatment with ALC and acute chasing stress.**

| Dependent variable | Effects | F-value | DF | P-value |
|---|---|---|---|---|
| Lipid peroxidation | Interaction | 14.70 | 3,24 | **0.0001** |
| (TBARS) | ALC | 14.39 | 3,24 | **0.0001** |
| | Stress | 8.80 | 1,24 | **0.0067** |
| Sulfhydryl | Interaction | 0.14 | 3,24 | 0.9339 |
| (SH) | ALC | 7.80 | 3,24 | **0.0008** |
| | Stress | 0.01 | 1,24 | 0.9289 |
| Non-protein thiol | Interaction | 2.73 | 3,24 | 0.0665 |
| (NPSH) | ALC | 3.63 | 3,24 | **0.0273** |
| | Stress | 6.35 | 1,24 | **0.0188** |
| Superoxide dismutase | Interaction | 5.46 | 3,23 | **0.0055** |
| (SOD) | ALC | 9.93 | 3,23 | **0.0004** |
| | Stress | 4.26 | 1,23 | 0.0504 |
| Catalase | Interaction | 0.13 | 3,21 | 0.9393 |
| (CAT) | ALC | 1.89 | 3,21 | 0.1626 |
| | Stress | 0.87 | 1,21 | 0.3606 |

**Notes.**
DF, degrees of freedom.
Significant effects ($p < 0.05$) are given in bold font.

Our study adds to a growing body of literature demonstrating the role of antioxidants in modulating behavior and oxidative homeostasis. The data presented here thus warrants further investigation of ALC as a potential agent in the treatment of neuropsychiatric illness. Its novel mechanism of action and good tolerability also subsidize the additional studies necessary to assess its therapeutic potential in clinical settings.

### Funding

This work was supported by Conselho Nacional de Desenvolvimento Científico e Tecnológico—Brazil (CNPq, No. 401162/2016-8 and 302800/2017-4). Ricieri Mocelin and Matheus Marcon are recipients of a fellowship from Coordenação de Aperfeiçoamento de Pessoal de Nível Superior (CAPES). The funders had no role in study design, data collection and analysis, decision to publish, or preparation of the manuscript.

### Grant Disclosures

The following grant information was disclosed by the authors:
Conselho Nacional de Desenvolvimento Científico e Tecnológico—Brazil: CNPq, No. 401162/2016-8, 302800/2017-4.
Coordenação de Aperfeiçoamento de Pessoal de Nível Superior (CAPES).

## Competing Interests

Angelo Piato is an Academic Editor for PeerJ.

## Author Contributions

- Lais Pancotto performed the experiments, authored or reviewed drafts of the paper, approved the final draft.
- Ricieri Mocelin conceived and designed the experiments, performed the experiments, analyzed the data, prepared figures and/or tables, authored or reviewed drafts of the paper, approved the final draft.
- Matheus Marcon conceived and designed the experiments, performed the experiments, approved the final draft.
- Ana P. Herrmann conceived and designed the experiments, analyzed the data, authored or reviewed drafts of the paper, approved the final draft.
- Angelo Piato conceived and designed the experiments, analyzed the data, contributed reagents/materials/analysis tools, prepared figures and/or tables, authored or reviewed drafts of the paper, approved the final draft.

## Animal Ethics

The following information was supplied relating to ethical approvals (i.e., approving body and any reference numbers):

All experiments were approved by the Ethics Committee of Universidade Federal do Rio Grande do Sul (#30992/2015).

## Data Availability

The raw data is provided as Data S1.

## Supplemental Information

Supplemental information for this article can be found online at http://dx.doi.org/10.7717/peerj.5309#supplemental-information.

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
