# Peer review of "Anxiolytic and anti-stress effects of acute administration of acetyl-L-carnitine in zebrafish"

_PeerJ, doi:10.7717/peerj.5309_

## Round 0.1 · original submission · Major Revisions

· Academic Editor

Major Revisions

Both reviewers require more detailed experimental procedures, specifically in what concerns, inter alia, the timing of feeding relative to the tests, the existence or absence of staggering of control and treated fish, the different length of the tests and the decision to evaluate the ACS test after NTT instad of LDT. Reviewer #2 further considers that the breadth of your conclusions requires more support and extensive additional experimentation. I agree that, at a minimum, the dramatic differences in lipid peroxidation require additional conformation with additional batches of fish.


Additional review-level observations by the editor:

In the description of the novel tank test, the reference to the original paper (E.D. Levin et al. Physiology & Behavior 90 (2007) 54–58) describing the method (and its similarity to the analysis of changes in anxiety/familiarity in rodents) is missing.

Maximino et al. have remarked, regarding the choice of material for the tank for the dark/light test that "The tank should be constructed
of matte acrylic that is as non-reflective as possible. Zebrafish are
a naturally shoaling species, and the use of reflective surfaces for
the tank can affect stimulus control, as the animals might display
social behavior in relation to their own reflections." You did, however, use glass tanks. Did you notice any indication that the previous worker's fears were warranted?

Reviewer 1 ·

Basic reporting

This manuscript is written well with only a few grammatical/stylistic issues.

The introduction is lacking content and is too short. There should be mention of the novel tank dive test and light/dark test and their use in zebrafish research. Why are the authors using zebrafish to test ALC? This must be clear early on before statements like those on line 204 that relate results to treating 'mental disorders' (line 204). More background is necessary linking the use of zebrafish as a medical model and details about why these tests are relevant. Has there been any research on ALC and fish?

The figures and tables were great.

Minor comments:
line 26 - "...and on excessive neuronal death." This does not make sense with the start of the sentence.
line 52 - a growing body of "evidence"
line 86, 87, 89 - change 'submitted to' to 'placed in'
line 105 - 'indicator'
line 143 - 'was mixed'
line 170 - 'Pearson'

Experimental design

It is interesting that fish were kept for only 15 days before being experimented upon. Why only 15 days?

Many facilities require a quarantine of 30 to 60 days (common duration of bacterial of parasitic illness) before experimentation.

The authors may want to consider this for future studies.

Were the experimental groups staggered with the control groups? Why were there less control fish than experimental fish in the first round of the NTT and LDT?

Why was the NTT done for 6 minutes and the LDT for 5 minutes? This should be explained in the manuscript.

Why was the ACS test done after the NTT and not the LDT? This should be explained in the manuscript.

In the result section (lines 177-178) it is stated that ALC significantly increased the time spent in the top. However, this is only true for 0.1 and 1.0 mg/L. This statement should be revised to represent the results. Moreover, why is there an opposite dose-response relationship here? 0.1 has a much greater effect than 1.0 and 10 mg/L. This needs to be addressed in the discussion.

Furthermore, in the NTT the 0.1 and 10 mg/L were significant but not the 1.0 mg/L. Why might this be? The authors also need to address this in the discussion.

Validity of the findings

The data is statistically sound, in my opinion. I liked the graphs (bars and individual data points) and diagrams.

Reviewer 2 ·

Basic reporting

There is a lot of room for improvement for the manuscript. See detailed comments below.

Experimental design

There is a lot of room for improvement for the manuscript. See detailed comments below.

Validity of the findings

There is a lot of room for improvement for the manuscript. See detailed comments below.

Additional comments

The manuscript by Pancotto et al. describes the anxiolytic properties of acetyl-l-carnitine (ALC) in a zebrafish model. Although the idea of the manuscript is interesting, the current manuscript should be extensively revised and additional experiments must be performed prior to this manuscript being considered for publication.

Major criticisms:
1. It would appear that the study was repeated on only one batch fish. Although multiple fish were used, the experiments must be repeated on at least two more independent cohorts of fish to ensure that there are no confounding factors that might be contributing to the reported effects.
2. The flake fish food diet must be properly characterized. This food is not commonly used with zebrafish (typically Zeigler's food with brine shrimp are used). The concern here is that this flake food may contain different levels of various antioxidants (if any) than other commercially available and more commonly used food. Since the antioxidant system is assessed, it is of major importance to analyze the diet for its antioxidant capacity, since this will have a major impact on oxidative stress parameters and the susceptibility of fish to oxidative stress and damage.
3. It is unclear how exactly the behavioral testing was performed. Please indicate whether the fish were fed before the testing and whether the treatment groups were alternated during the tests. In other words, was the control group analyzed first followed by the exposed groups, or was the order alternated. This is crucial information. The behavioral testing was performed between 9 am and 4 pm, based on personal experiences with Novel tank test, fish that are tested before lunch perform much better than those tested post lunch, because they are getting hungry, so it is crucial to reduce this effect by feeding the fish in the morning and alternating the treatment groups.
4. It is unclear whether the TBARS assay included butylated hydroxy toluene (BHT) in the incubation mixture and whether or not butanol extraction was performed. Based on the current description, these aspects are missing. Not including BHT and not performing butanol extraction will result in overestimation of TBARS.
5. The lipid peroxidation in the brains of control stressed fish is ~2 fold higher than non-stressed fish. This is a very striking result, considering that the fish were stressed only for a few minutes. It would be essential to consider the aforementioned comments #2-4 and repeat these measurements with independent fish cohorts to ensure the accuracy.
6. The study focuses on anxiety and stress, yet there was no attempt to assess cortisol levels to verify the hypothesis that the observed effects of ALC are actually due to lower stress. Cortisol levels have been measured previously in zebrafish, both as whole fish lipid extracts and in plasma (e.g. Filby et al., 2010, Physiol Behav; Fuzzen et al., 2011, Gen Comp Endocrinol; Massarsky et al., 2014, Sci Tot Environ). The cortisol data must be included to validate the reported results. This is especially critical since the behavioral tests are based upon locomotor activity, so in theory it is possible that the behavioral effects are due to bioenergetic changes rather than neuronal. Inclusion of cortisol data could support the neuronal changes.
7. The conclusion states that the study demonstrates that ALC can modulate neurochemical homeostasis. This conclusion cannot be reached based upon the performed experiments since neurotransmitters were not assessed. The authors should either remove this statement or attempt to measure neurotransmitters in the brain, which has been done recently in the zebrafish brain using ELISA kits (e.g. Massarsky et al., 2018, NeuroToxicology) or HPLC (Chatterjee and Gerlai, 2009, Behav Brain Res).
8. Several recent studies demonstrate the behavioral effects in zebrafish can be sex-dependent (e.g. Weber et al., 2015, J Toxicol Environ Health; Massarsky et al., 2018, NeuroToxicology). Therefore, to get a better idea of what ALC is doing in zebrafish, the results must be separated by sex. It is possible that males are more sensitive than females or vice versa. Separation by sex could potentially reduce the huge variation that is seen in the behavioral endpoints.
9. The study tries to relate the ALC supplementation (its antioxidant properties) as a potential treatment for anxiety in humans. However, there are already a number of commercially available antioxidants as well as multiple antioxidants that are present in the diet that humans are exposed to. Therefore, the aim of the study is not entirely clear with that regards. If the authors are trying to convince the audience that ALC is more efficient than other compounds (e.g. N-acetyl cysteine), than a proper experimental comparison must be made between ALC and other compounds to validate the efficacy and relevance.

Minor criticisms:
1. Running title: 'Acute ALC' should be followed by 'exposure' to make more sense.
2. Keywords: 'acute' can be removed from the first keyword. 'acute stress' should be 'oxidative stress'.
3. Abstract:
- lines 24-27 - the end of the sentence 'and on excessive neuronal cell death' does not make sense grammatically. The sentence should be revised accordingly.
- line 33 - 'this drug' should be replaced by 'ALC'. The obtained chemical is not actually a pharmacological substance and should not be referred to as 'drug'.
- lines 35-36 - there is practically no mechanistic data in the current manuscript. Therefore 'its novel mechanism of action' should be removed.
4. Introduction:
- The introduction needs to be revised to include more pertinent background information to set a better and more logical stage for the study. The topic of oxidative stress should be clarified; oxidative stress is a complex issue that has multiple stages initially there is no response, followed by induction of antioxidant genes, followed by oxidative damage (if antioxidant system fails).
- lines 48-49 - the portion that starts with 'it acted as a rapid...' should be revised.
- line 52 - 'evidences' should be 'evidence'.
- line 55 - 'organism' should be 'organism's'
- line 57 - there are a lot of antioxidant compounds that have been evaluated in clinical studies. Please provide the specific examples that are actually relevant to the current study.
5. Materials and methods:
- The title of the section should be 'Materials and methods' not 'Material and methods'.
- lines 72-72 - please specify the housing conditions rather than referring to previous study.
- line 74 - why was the temperature set at 26 degrees? The optimal temperature for zebrafish is actually 28 degrees.
- line 77 - please provide more details on the fish diet since the mentioned food is not commonly used in zebrafish studies. See also comment #2 in major criticisms.
- line 84 - was there a control group? the current description of treatment groups only includes the 0.1, 1, and 10 mg/L ALC.
- line 84 - how were the ALC concentrations chosen? Need to justify why these concentrations and how they are relevant to humans.
- line 88 - how was the 'chasing stress' chosen? Has this stress method been validated in previous studies? Need to justify.
- lines 90-91 - according to the provided description the control group was not subjected to chasing stress. Why this was the case?
- line 94 - need to mention if the groups were alternated. See comment #3 in major criticisms.
- line 129 - why were the brains pooled? The zebrafish brain is typically ~5 mg, which is more than enough tissue to measure TBARS and antioxidant levels. please justify.
- Please provide more details on TBARS assay. See comment #4 in major criticisms.
- line 149 - why were not glutathione levels measured? GSH is a major antioxidant and has been one of the commonly used oxidative biomarkers in fish species.
- line 158 - SOD and CAT are rarely affected by oxidative stress conditions in fish. Why were not GR and GPx activities measured instead (or in addition). GR is a much more sensitive biomarker than SOD/CAT.
6. Results:
- The results section should be better organized and separated into sections (e.g. behavior, antioxidants, oxidative damage, etc) for better clarity.
7. Discussion:
- The discussion is somewhat general and does not properly explain why the 0.1 mg/L ALC was more effective than 10 mg/L. Also, are the obtained results relevant to human patients? How is the antioxidant properties of ALC compare to other common antioxidants? Given the availability of numerous antioxidants on the market, how can inclusion of ALC supplements be relevant for improving outcomes?
8. Conclusion:
- The authors mention 'neurochemical homeostasis', but neurochemicals were not actually assessed in this study. See comment #7 in major criticisms.

---

## Round 0.2 · Major Revisions

· Academic Editor

Major Revisions

Please pay attention to all the issues raised by our reviewerw. I do concur with reviewer #2 regarding the need for additional batches of fish, and I will not be able to accept this manuscript until sufficient replication is performed. Do ensure that lipid peroxidation assays include butylated hydroxytoluene, as pointed out by the reviewer.
While I am more sensitive to your underfunding issues than reviewer #2, I agree that cortisol levels would provide a much less ambiguous signal and seem to be absolutely required to specifically to rule out the possibility of the changes being due to effects of acetyl-L-carnitine on locomotion, vision and perception.

Reviewer 1 ·

Basic reporting

The authors have addressed most of my issues.

Experimental design

The authors have addressed most of my issues.

Validity of the findings

The authors have addressed most of my issues.

Additional comments

The authors have attempted to revise the manuscript, however, the following still needs to be addressed.

Zebrafish (Danio rerio, F. Hamilton 1822) have emerged as a suitable model organism in the field of mental disorders
-This is an overstatement and should be revised or removed. The line following this is more accurate.

The novel tank test is used to evaluate locomotor and exploratory activities and corresponds to the open-field test in rodents, whereas the light/dark test evaluates anxiety based on the innate preference of zebrafish for dark instead of bright spaces

-this is still not correct. An open field test in zebrafish corresponds to an open field test in rodents. The novel tank diving test is based on an anti-predatory defense mechanism that causes fish to swim at the bottom of the tank. As anxiety-like behaviour is decreased they swim higher in the water column. This is an excellent test of anxiety-like behaviour and does not correspond to the open-field test in rodents.
-secondly, the light/dark test evaluates anxiety based on the innate preference of adult zebrafish to prefer dark over light areas. Larval zebrafish have been shown to demonstrate a light preference.

Reviewer 2 ·

Basic reporting

- There are a few awkward sentences in the manuscript. I would recommend that the authors ask a native English speaker to go through the manuscript.
- The context should be further improved.

Experimental design

- The investigation is not as rigorous as one would hope.
- Some methods are incorrect.

Validity of the findings

- Data is not robust to support the conclusions.
- The anxiolytic effects are speculative and are not supported by biochemical data.
- Insufficient replication of the experiments.

Additional comments

The revised ms by Pancotto et al. is slightly improved from the original version, but unfortunately several of the major issues that were raised by reviewer #2 were not adequately addressed. I regret to inform that I cannot recommend this manuscript for acceptance at this point.

Major concerns:

1. According to the authors, the experiments were repeated on only 2 batches of animals. This is an insufficient replication number. All experiments, no matter how tedious and frustrating, must be repeated 3 independent times!
a. This is especially important for behavioral assays. Our behavioral battery consists of 4 different behavioral tests. We conduct these tests using at least 3 independent cohorts of fish with 10 males and 10 females per treatment. So, we have at least 30 fish of each sex by after testing with 3 cohorts of fish. And even then the variation across fish and cohorts is quite large. Moreover, if cohort 1 showed a particular trend, the next cohort may or may not show the same trend. It is extremely important to test behavioral endpoints (just like any other endpoint) with at least 3 independent cohorts!
b. The authors claim that space was an issue and so they could not repeat the experiments with 3 cohorts. I don't find this excuse appropriate, especially since most fish were sacrificed after the experiments described in the manuscript, which would have created space to house additional fish and perform the experiments one more time.
c. Biochemical parameters should also be repeated with fish from 3 independent cohorts to make sure that the reported results are accurate.

2. The authors claim that there were not significant sex differences and therefore the data were pooled together. Although there may not be sex differences, the data MUST be separated by sex, to provide a more accurate picture. This applies to both behavioral and biochemical parameters. Besides, sex differences may become more apparent if the experiments are repeated with another cohort of fish. Also, some tissues were pooled together for analysis, were males pooled with males and females with females, or was it done at random?
In our experience, there are always behavioral differences related to sex. Females appear much more susceptible to handling/netting stress and take longer to recover during the behavioral tests. Therefore, ALC may have differential potency in males and females. Please separate all results by sex (even if no significant difference are apparent) and explain in the manuscript that sex was considered as a factor!

3. The reported anxiolytic effects must be supported by biochemical data! Although the behavioral data suggests that ALC has anxiolytic properties, these results are highly speculative. The behavioral tests depend a lot on locomotion and vision, and so any effects of these two parameters will skew the results. For example, perhaps ALC somehow interferes with vision in the light and dark test, and therefore the fish are unable to detect the lighting difference as efficiently as the control fish.
It was previously suggest to examine the cortisol levels, which would be a more direct and stress-related biochemical parameter that would offer a less speculative view. The authors response to this suggestion was that there are insufficient funds to perform such studies. I find this excuse inappropriate. All labs, not matter how big, have financial considerations. An ELISA assay for cortisol would cost around $500 (which is less than the publication fee of Peer J). Alternatively, the authors can form collaborations with labs that routinely measure cortisol levels in zebrafish. Without cortisol data the authors would have to remove anxiolytic/stress from the title and other sections of the manuscript.

4. The lipid peroxidation remains an issue! I find it highly unbelievable that 10 min stress will result in 1.5 fold increase in lipid peroxidation in the brain. The TBARS assay specifically has been heavily criticized in the literature because it is not very specific. As a person who performed this assay, I would not totally discredit this assay; however, every precaution must be taken to minimize the 'unspecificity' of this test. inclusion of BHT in the reaction mixture is a must, as it reduced the extent of lipid damage during 30 min at boiling temperature. Without BHT, the lipid peroxidation will be overestimated, which I suspect is the case in the current manuscript. Moreover, butanol extraction must be included. This is especially of concern for tissues because of presence of blood/hemoglobin that absorbs at similar wavelength as MDA. Without butanol extraction, the lipid peroxidation will be overestimated.
The fact in previous papers the TBARS assay was performed without BHT or butanol extraction is not an appropriate excuse not to perform this assay correctly, especially with such surprising results (i.e. 1.5 fold increase in lipid peroxidation after 10 min of stress). The TBARS assay must be repeated correctly, if the authors want the readers to believe that their results are accurate.

Minor concerns:

1. There are a few awkward sentences throughout the manuscript. I would recommend the authors to have their manuscript read by a native English speaker.
2. I think additional experimental details would be great, especially specifying the number of times the experiments were independently repeated, the sex composition of fish that are being used for various assays, etc.

---

## Round 0.3 · Minor Revisions

· Academic Editor

Minor Revisions

I am sorry for the delay in this response but I the new reviewer who agreed to evaluate your manuscript has unfortunately failed to provide a review to help sort out the differences between you and reviewer#2.

Although the request of additional experiments is in line with journal policies regarding the presentation of robust data, I recognize the cogency of your rebuttal arguments regarding the replication request. As a way to address reviewer#2's concerns regarding your TBARS data, I would like you to provide a short description of why you believe your TBARS results are robust in spite of the lack of BHT, or alternatively discuss the possible limitations of your assay and the implications thereof on your data interpretation.

---

## Round 0.4 · accepted · Accept

· Academic Editor

Accept

Although I think the arguments you provide in the rebuttal letter are better worded than the ones you decided to use in the manuscript, I think the paper can be accepted as is.